# Circuit Explained: How Does a Transformer Perform Compositional Generalization

## Abstract

Compositional generalization — the systematic combination of known components into novel structures — is fundamental to flexible human cognition, yet the mechanisms that enable it in neural networks remain poorly understood in both machine learning and cognitive science. Lake & Baroni (2023) showed that a compact encoder-decoder transformer can achieve simple forms of compositional generalization in a sequence arithmetic task. In this work, we identify and mechanistically interpret the circuit responsible for this behavior in such a model. Using causal ablations, we isolate the circuit and show that this understanding enables precise activation edits to steer the model's outputs predictably. We find that the circuit performs function composition without encoding the specific semantics of any given function — instead, it leverages a disentangled representation of token position and identity to apply a general token remapping rule across an entire family of functions. Our findings advance the understanding of how compositionality can emerge in transformers and offer testable hypotheses for similar mechanisms in other architectures and compositional tasks. Code will be released after double-blind review.

**Keywords:** Transformer; Mechanistic Interpretability; Compositionality

## 1 Introduction

Humans excel at compositional generalization—the ability to combine known components to solve novel problems. For example, once we learn the concept of "swap," we can apply it to any two objects, regardless of their identity. This capacity for relational abstraction appears early: even infants generalize rules like same-different or before-after to unfamiliar contexts (Gentner, 1983; Holyoak & Thagard, 1994; Marcus et al., 1999). While cognitive science has long studied this ability, its neural basis remains unknown.

In parallel, compositional generalization has posed a longstanding challenge in machine learning. Building flexible, general-purpose systems requires models to go beyond memorization and exhibit systematic generalization. Classic critiques, such as those by Fodor (1979), argued that connectionist models lack the structure required for compositionality. While this concern shaped decades of skepticism, recent advances with transformer-based models (Vaswani et al., 2017)—and large-scale training paradigms (Kaplan et al., 2020; Bubeck et al., 2023) have provided substantial counterexamples.

In recent work, Lake & Baroni (2023) demonstrated that a compact encoder-decoder transformer can achieve human-like compositional generalization in a symbolic sequence arithmetic task. Here, we leverage the small scale of this model and provide an end-to-end mechanistic interpretation of how it solves this compositional generalization task. We trace the model's behavior to a minimal, interpretable circuit and reverse-engineer the attention dynamics into a human-readable algorithm. We validate the circuit's role by predictably steering the model's output.

This study reveals two core principles underlying the model's success:

1. The model relies on disentangled representations of position (slot) and token (content) embeddings.

2. *Functions* operate on disentangled position embeddings, allowing generalized transformations of arbitrary token identities.

Previous studies have identified *function vectors* in transformers — semantically meaningful activations that represent abstract transformations (Todd et al., 2023; Stoehr et al., 2024). In contrast, we uncover a different mechanism: the circuit applies functions to new arguments by remapping the output pattern through token-routing, without encoding the actual semantics of the functions themselves. While our model and task are intentionally specific, the mechanism we identify may underlie similar forms of compositional generalization that involve sequence remapping, such as word completion (Olsson et al., 2022), object identification (Wang et al., 2022), and code completion (Husein et al., 2025).

We discuss **Related Work** on *Transformer Circuit Interpretation* and *Compositional Generalization in Transformers* in the Appendix.

## 2 Experimental Setup

Our experimental setup involves a synthetic function composition task (Figure 1) designed to probe *compositional generalization* in a compact Transformer (Lake & Baroni, 2023). We outline the task structure, the Transformer basics (including attention mechanisms), and the training protocol.

### 2.1 Task Structure

Each episode consists of a **Support Set** and a **Question** (Figure 1):

**Support Set:**

Specifies (i) how the *Primitives* are symbol-to-color mappings (e.g., B maps to blue, written as [B] = blue, with [ ] as an interpretation function for translating inputs to outputs; or D maps to pink, written as [D] = pink). The Support Set also specifies (ii) *Functions* as symbolic operations that take primitives as arguments, and calls the interpretation function on those arguments in a new order. For instance, the function S might take two arguments and evaluate them in the order of (e.g., [B S D] = [D][B][D] = pink blue pink. The 2nd token is always the *function* token; the 1st and 3rd tokens are always the *argument* tokens to the *function*. The length of *function* output ranges from 2 to 5. *Function* generation rule is further described in A.3.

**Question:**

Presents a new composition of primitives and functions defined in the Support Set.

The model generates answers to the **Question** as token sequences emitted from the decoder, with a

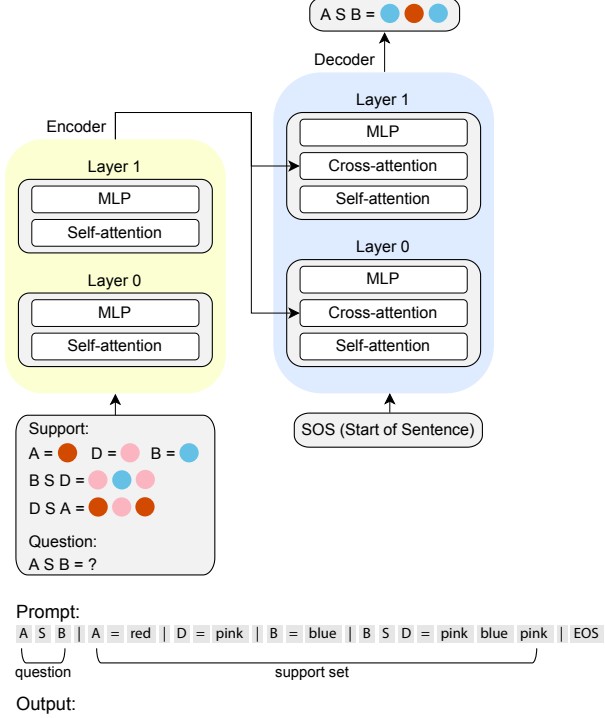

Figure 1: Top, schematic of the transformer model and task. Bottom, the prompt and output format for the compositional generalization task.

SOS (start of sentence) token as the first input to the decoder and an EOS (end of sentence) marking the end of the emission. The model operates strictly via in-context learning—weights remain frozen during inference, and test episodes are disjoint from training data. The model must infer latent variable bindings (primitives and functions) from the **Support Set** and dynamically compose these bindings to solve the novel **Question**.

### 2.2  Model

#### 2.2.1  Transformer Basics

Our transformer uses an encoder-decoder architecture that involves two types of attentions:

- **Self-Attention:** Captures within-sequence interactions. The token embedding matrix $X \in \mathbb{R}^{n_{\text{input}} \times d_{\text{model}}}$ is projected into Queries, Keys, and Values:

$$Q = XW_Q, \quad K = XW_K, \quad V = XW_V,$$

  where $W_Q, W_K, W_V \in \mathbb{R}^{d_{\text{model}} \times d_{\text{head}}}$ are learnable weight matrices.

- **Cross-Attention:** Enables the decoder to attend to encoder outputs. Here, the Queries ($Q$) come from the *decoder* tokens, while the Keys ($K$) and Values ($V$) come from the *encoder* tokens. We denote the different tokens used to compute $Q$ $K$ $V$ as $X_Q$ $X_K$ $X_V$.

The attention mechanism operates through two separate circuits on embedding $X \in \mathbb{R}^{n_{\text{input}} \times d_{\text{model}}}$ for each attention head:

- **QK Circuit** ($W_Q W_K^\top$): Determines from *where* information flows to each token by computing attention scores between token pairs, with higher scores indicating stronger token-to-token relationships:

$$\text{Attention}(Q, K) = \text{softmax}\left( \frac{X_Q W_Q (X_K W_K)^\top}{\sqrt{d_{\text{head}}}} \right) \in \mathbb{R}^{n_{\text{query}} \times n_{\text{key}}},$$

  where *softmax* is applied along the Key dimension and independently for each head.

- **OV Circuit** ($W_V W_O$): Controls *what* information gets written to each token position. Combined with the **QK Circuit**, this produces the output of the attention head:

$$Z = \text{Attention}(Q, K) \cdot X_V W_V W_O \in \mathbb{R}^{n_{\text{query}} \times d_{\text{model}}},$$

  where $W_O \in \mathbb{R}^{d_{\text{head}} \times d_{\text{model}}}$ is a learnable weight matrix.

Following attention blocks, each transformer layer includes a position-wise **MLP block**, which applies a non-linear transformation to each token embedding independently. However, prior work by Wang et al. (2022) demonstrated that MLPs are less relevant for induction tasks. Therefore, our circuit analysis focuses exclusively on attention heads.

#### 2.2.2  Model Training

We adopt an encoder-decoder Transformer with **2 layers** in the encoder and **2 layers** in the decoder (Figure 1) with each layer containing **8 attention heads**. Further model details appear in the Appendix. We have also experimented with different model hyperparameters, see Appendix A.6.

We train on 10,000 such episodes for 50 epochs and evaluate on 2,000 test (held-out) episodes. The model achieves **94%** accuracy on this test set, indicating strong compositional generalization capabilities. In the test set, primitive assignments and function definitions are conjunctively different from those in the training set (i.e., some primitives or some functions might be in the training set, but not the whole combination of them), preventing a memorization strategy. Please refer to the Appendix for additional details.

## 3  Results

We first outline the high-level algorithm the model appears to implement. We then describe our circuit discovery process, using causal methods to isolate the attention heads driving compositional behavior. Finally, we validate the mechanism through targeted interventions that predictably steer model outputs. Throughout, we use 1-indexing (count from 1) for tokens.

### 3.1 The High-Level Algorithm

The overall algorithm the model implements can be described in terms of three logical steps (Figure 2).

1. **Unbind**: The *function* output in the Support is coded in terms of an array of positions associated with the *argument* tokens.

2. **Rebind**: The new *argument* tokens in the Question bind with the original *argument* positions in the Support forming new position-token pairs.

3. **Produce**: Reapply the function coded in terms of positions to the new position-token pairs to form an array of output tokens.

In essence, the model uses token-independent positions as a pointer system that through **unbinding** and **rebinding** can generalize to different token identities.

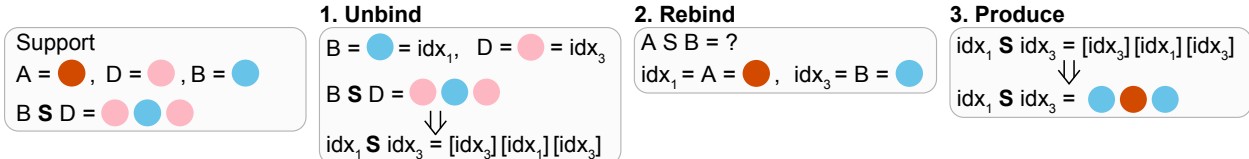

Figure 2: Schematic of the compositional algorithm. For **Unbind**, in 'B S D', B is the `1st` token ($idx_1$), and 'B=blue', so 'B=blue=$idx_1$'; similar for 'D'. For **Rebind**, in 'A S B', A is the `1st` token ($idx_1$), and 'A=red', so '$idx_1$=A=red'; similar for 'B'.

### 3.2 Transformer Solution with Attention Operations

Next, we map each step to specific attention heads and walk through the attention operations with a guidance episode in Figure 3.

Nomenclature: given a function definition in the prompt (e.g., '...| B S D=pink blue pink|...'), function Left-Hand-Side (LHS) refers to tokens on the left (e.g., 'B S D'); function Right-Hand-Side (RHS) refers to tokens on the right (e.g., 'pink blue pink'). Index-on-LHS/RHS refers to the *relative* position of tokens on the LHS/RHS, (e.g., 'B=1st, pink=1st'). Similarly, index-in-question refers to a token's *relative* position in the *Question*. The attention heads are named by their interpreted roles, which are detailed in the **Circuit Discovery** section.

**Step 1: Unbind**

- **Primitive- and Function-Retrieval Heads** (Figure 3a): Color tokens on the function RHS (`pink`) attend to their associated primitive tokens on the function LHS (`D`), inheriting the latter's index-on-LHS (`3rd`). The heads are detailed in Figure 9.

**Step 2: Rebind**

- **Question-Broadcast Head** (Fig. 3b): Primitive input tokens in the Support (e.g., `B`) attend to the same primitive tokens in the Question (`B`), inheriting the latter's index-in-question (`3rd`). The head is detailed in Figure 6b.

- **Primitive-Pairing Head** (Fig. 3c): Color tokens (`blue`) attend to their associated primitive tokens (`B`), inheriting the latter's index-in-question (`3rd`). The head is detailed in Figure 6a.

**Step 3: Produce**

- **RHS-Scanner Head** (Fig. 3d): The `1st` token in the Decoder (`SOS`) attends to the `1st` tokens on the function Right Hand Side (RHS) (`pink`), inheriting the latter's former-inherited index-on-LHS (`3rd`). The head is detailed in Figure 8.

148 • **Output Head** (Fig. 3e): `SOS` token (with inherited index-on-LHS=`3rd`) attends to color tokens
149 (`blue`) with the same index-in-question (`3rd`), inheriting the latter's token identity (`blue`), and
150 generates the next prediction (`blue`). The head is detailed in Figure 5. Then the `2nd` token in
151 the Decoder (`blue`) repeats the operation with the RHS-Scanner Head until completion of function
152 RHS.

153 Together, these heads form a modular circuit that performs compositional generalization via pointer-based
154 binding, mirroring symbolic execution through argument extraction, rebinding, and output generation.

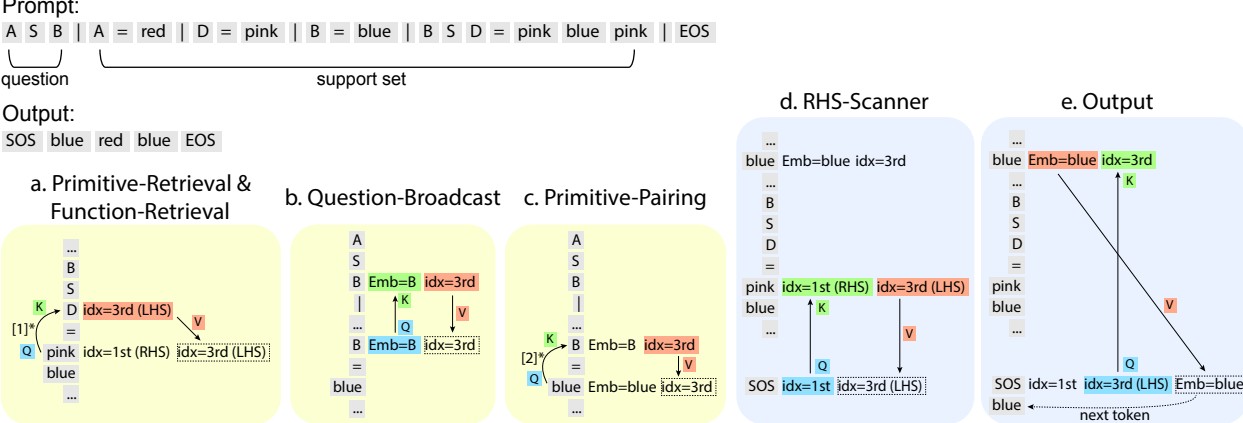

Figure 3: Summary of circuit for compositional generalization. Top, the example episode's input and output. For a-e, the yellow boxes indicate self-attention heads and the blue boxes indicate cross-attention heads. Titles refer to the functional attention heads that execute the steps (details in Circuit Discovery section). We unfold all relevant information superimposed in tokens' embeddings and highlight their roles in attention operations. [1]*, the $QK$ alignment discussed in Primitive-Retrieval Head section. [2]*, the $QK$ alignment discussed in Primitive-Pairing Head section.

## 3.3 Circuit Discovery

156 **Nomenclature.** For attention heads, *Enc-self-0.5* stands for *Encoder, self-attention, layer 0, head 5*;
157 similarly, *Dec-cross-1.5* stands for *Decoder, cross-attention, layer 1, head 5*.

158 **Minimal Sufficient Circuit.** Circuit interpretation studies often find that a given behavior is supported
159 by multiple redundant or compensating circuits. We observe this redundancy in our model as well (Lieberum
160 et al., 2023; Conmy et al., 2023). To facilitate clear mechanistic interpretation, we focus our analysis on
161 a *minimal sufficient circuit*, defined by two criteria: (1) this circuit alone can account for the behavior of
162 interest, and (2) the circuit is pruned based on each head's contribution until all included heads perform
163 functionally unique roles.

164 We first present the full diagram of the key attention heads to illustrate their relationships (Figure 4). We
165 then describe the detailed role of each head with reference to this diagram in the corresponding sections that
166 follow. The circuit is also replicated in another model with different hyperparameters (Appendix A.6).

### 3.3.1 Output Head (Dec-cross-1.5; Figure 5)

168 We discovered the model's circuit backwards from the unembedding layer using *logit attribution* (nostal-
169 gebraist, 2020), which measures each decoder attention head's linear contribution to the final token logits
170 (adjusted by the decoder's output *LayerNorm*). We identified **Dec-cross-1.5** (decoder cross attention layer
171 1 head 5) as the primary contributor (Figure 5a).

172 Dec-cross-1.5's $Q$ tokens always attend to the $K$ tokens from the Encoder that are the *next* predicted ones.
173 For example, in Figure 5b, the `SOS` token attends to instances of `red` in the Support Set, which is indeed the

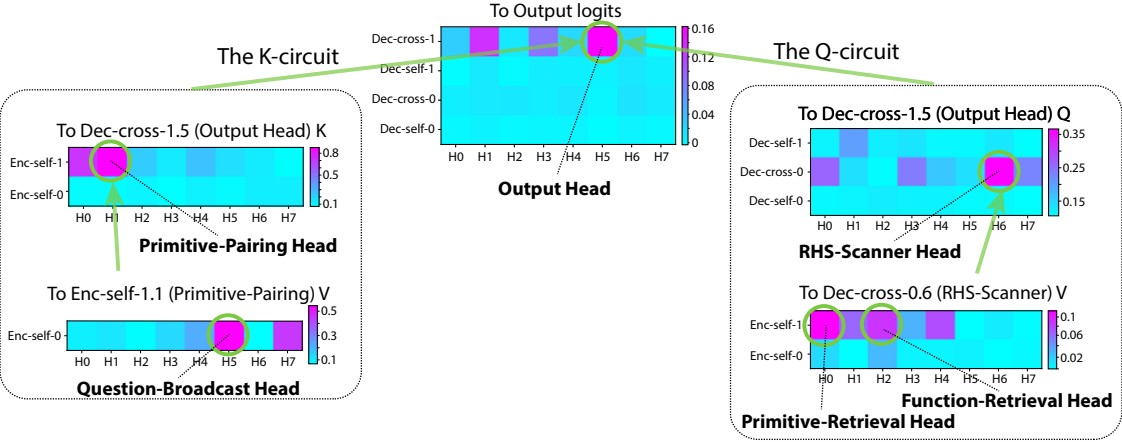

Figure 4: Circuit diagram of the key attention heads. Green circles indicate attention heads that contribute most significantly to downstream nodes. Green arrows denote the flow of contributions from upstream nodes to each attention head. The main sub-circuits highlighted are the *K*-circuit and *Q*-circuit leading to the Output Head.

correct next output prediction. This attention accuracy (i.e., max-attended token being the next-emitted token) of Dec-cross-1.5 remains above 90% for the first three tokens in the responses across all test episodes (Figure 5c), with Dec-cross-1.1 and -1.3 partially compensating beyond that point.

These observations suggest that Dec-cross-1.5's *OV* circuit feeds token identities directly to the decoder unembedding layer (output layer). Specifically, we observe that the output of the *OV* circuit, $XW_vW_o$, align closely (strong inner product) with the unembedding vectors of the corresponding tokens (Figure 5d). Hence, we designate Dec-cross-1.5 as the *Output Head* (while Dec-cross-1.1 and -1.3 perform similar but less dominant roles).

Next, we show how the Output Head identifies the correct token through $QK$ interactions.

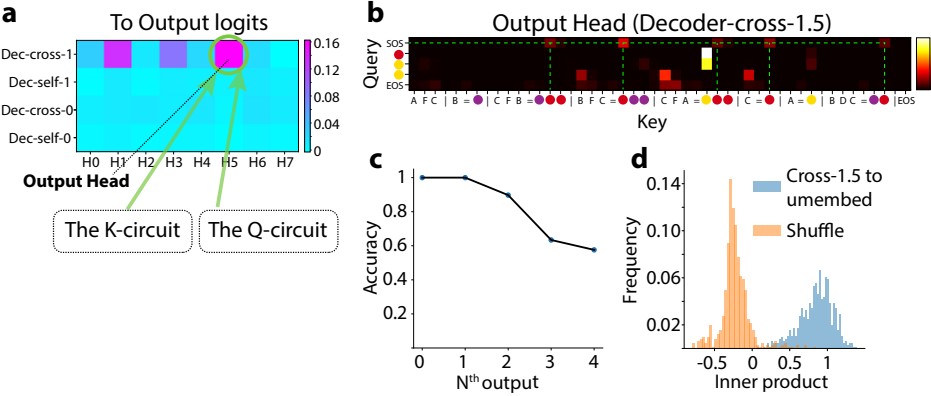

Figure 5: (a) Logit contributions of each decoder head to the logits of correct tokens (fraction to total logits). (b) Attention pattern of Dec-cross-1.5. (c) For Dec-cross-1.5, the percentage of attention focused on the next predicted token. (d) For Dec-cross-1.5, alignment (inner product) between its *OV* output (e.g., $x_{red}W_vW_o$) and the corresponding unembedding vector (e.g., Unemb$_{red}$). We estimated the null distribution by randomly sampling unembedding vectors.

### 3.3.2  The K-Circuit to the Output Head

We first determine which encoder heads critically feed into the Output Head's *K*. To do this, we performed *path-patching* (Wang et al., 2022) by ablating all but one single encoder head (keep-only-one-head) and then

measuring how much of Output Head's $QK$ behavior (*i.e.*, attention accuracy) remained. During these experiments, Output Head's $Q$ were *frozen* using clean-run activations. Here we report patching results with mean-ablation (qualitative similar to random-sample ablation) (details in Appendix).

Through this process, we identified **Enc-self-1.1** and **Enc-self-0.5** as the primary contributors to Output Head's $K$, acting in a sequential chain (Figure 6a). Next, we show how they sequentially encode symbols' index-in-question critical for the $QK$ alignment.

**Primitive-Pairing Head (Enc-self-1.1; Figure 6b)** This head exhibits a distinct attention pattern that pairs each color token with its associated primitive symbol token (e.g., in the Support Set, all instances of `red` attend to `C`). In other words, Enc-self-1.1 relays information (described below, as computed by e.g., **Enc-self-0.5**) from the primitive symbols to their corresponding color tokens via its $QK$ circuit. Hence, we call Enc-self-1.1 the *Primitive-Pairing Head.*

To investigate which upstream heads feed into the $OV$ circuit of the Primitive-Pairing Head, we applied a sequential variant of path-patching, isolating the chain:

Upstream heads (e.g. Enc-self-0.5) $\longrightarrow$ Primitive-Pairing Head ($V$) $\longrightarrow$ Output Head ($K$),

while mean-ablating all other direct paths to Output Head's $K$. We identified **Enc-self-0.5** as the most contributing node (Figure 6c).

**Question-Broadcast Head (Enc-self-0.5; Figure 6c)** All input symbol in the Support Set attend to their copies in the input Question. In other words, Enc-self-0.5 broadcasts question-related information (including token identity and position) across symbols in the Support Set (henceforth the Question-Broadcast Head). We hypothesize that the primitive symbols' index-in-question is the critical information passed from the Question-Broadcast Head's $Z$ through the Primitive-Pairing Head's $Z$ and lastly into the Output Head's $K$.

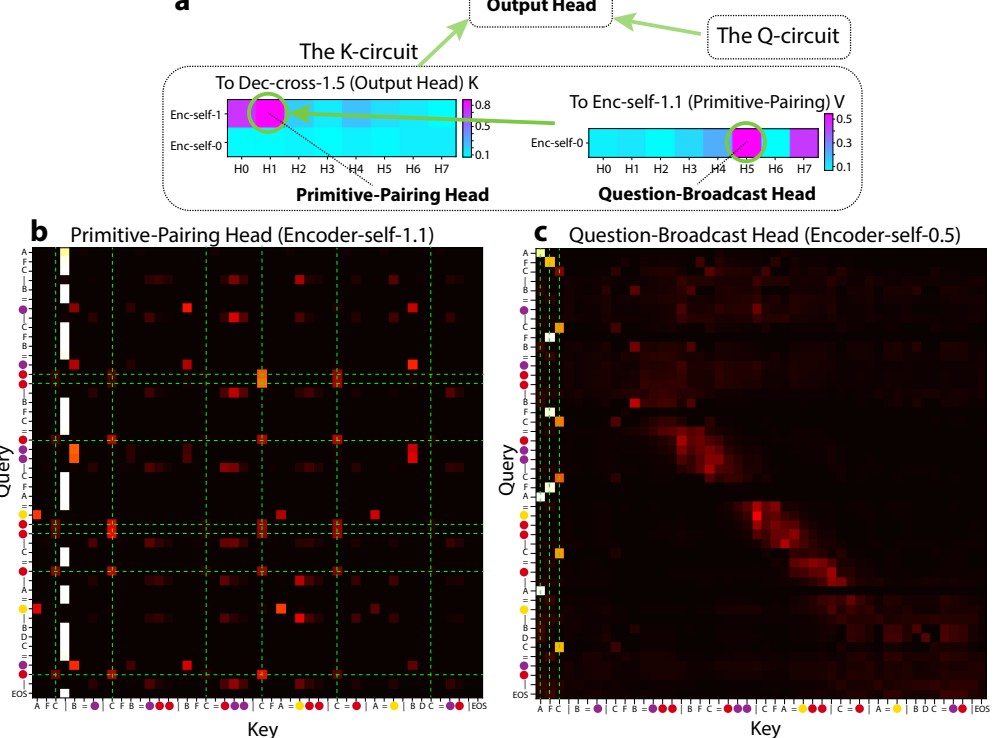

Figure 6: (a) Top, contributions to Output Head's performance (percentage of attention on the correct next token) via $K$. Bottom, attention pattern of Enc-self-1.1. (b) Top, contributions to the Output Head's performance through the Primitive-Pairing Head's $V$. Bottom, attention pattern of Enc-self-0.5.

**Index-In-Question Tracing** To validate this hypothesis, we examined the Question-Broadcast Head's $Z$ for each *primitive-symbol* token. We reduced these outputs to two principal components and colored each point by its index-in-question. As illustrated in Figure 7a, the Question-Broadcast Head's $Z$ exhibit clear clustering, indicating that the index-in-question is robustly encoded at this stage (quantified by the $R^2$ score, i.e., the amount of variance explained by index identity, details in Appendix). We further confirmed that the Primitive-Pairing Head's $Z$ preserves index-in-question (Figure 7b) and that the resulting Output Head's $K$ also reflect the same clustering (Figure 7c).

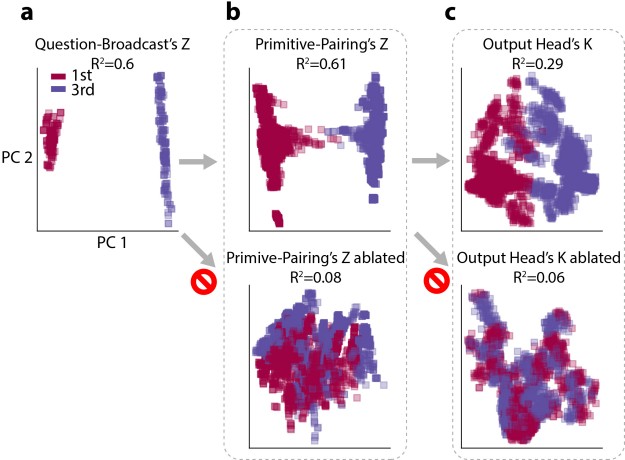

Figure 7: Principal Components Analysis (PCA) of token embeddings, colored by their associated index-in-question. Concretely, for a prompt like 'A S B | A=red | B=blue | ...', in (a), points are the $Z$ of 'A' and 'B' in the Support (`A` labeled `1st`, `B` labeled `3rd`); in (b), points are the $Z$ of 'red' and 'blue' in the Support (`red` labeled `1st`, `blue` labeled `3rd`); in (c), points are the $K$ of 'red' and 'blue' in the Support (`red` labeled `1st`, `blue` labeled `3rd`). The distinct clusters suggest strong index information. $R^2$ score quantifies the percentage of total variance explained by the index identity.

**Causal Ablation** Finally, we verified that this circuit indeed causally propagates index-in-question. Ablating the Question-Broadcast Head's $Z$ (together with the similarly functioning Enc-self-0.7) obliterates the clustering in the Primitive-Pairing Head's $Z$; ablating the Primitive-Pairing Head's $Z$ (together with similarly functioning Enc-self-1.0) disrupts the clustering in the Output Head's $K$ (Figure 7). We therefore conclude that the Question-Broadcast Head, the Primitive-Pairing Head and heads with similar functions form a crucial $K$-circuit pathway, passing index-in-question information from primitive tokens to their associated color tokens in the Output Head's $K$.

### 3.3.3 The Q-Circuit to the Output Head

Having established the role of the $K$-circuit, we next investigate where its $Q$ originates. We again relied on *sequential path-patching* to pinpoint which decoder heads ultimately provide the Output Head's $Q$. We identified **Dec-cross-0.6** as the main conduit for the $Q$ values of the Output Head. Enc-self-1.0 and -1.2 supply positional embeddings that enable the decoder to track primitive symbol's index-on-LHS, thereby completing the $QK$ alignment for correct predictions (Figure 4).

**RHS-Scanner Head (Dec-cross-0.6; Figure 8)** We identify **Dec-cross-0.6** as the dominant contributor to the the Output Head's $Q$ (Figure 8a). Analyzing Dec-cross-0.6's attention patterns reveals that each $Q$ token (from Decoder in the cross-attention) sequentially attends to the color tokens (in the Support Set) on the function's RHS (Figure 8b). For example, the first Decoder token (`SOS`) attends to the first RHS tokens (`purple`, `red`, `yellow`), and the second query token (`red`) attends to the second RHS tokens (`red`, `purple`, `red`), and so on. This iterative scanning mechanism enables the decoder to reconstruct the transformation defined by the function. Hence we call Dec-cross-0.6 the RHS-Scanner Head.

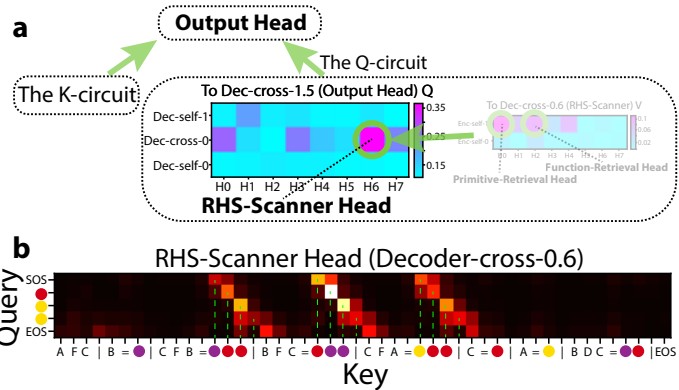

Figure 8: (a) Contribution to Output Head's performance via $Q$.(b) Attention pattern of Dec-cross-0.6.

**Primitive-Retrieval Head (Enc-self-1.0; Figure 9b) and Function-Retrieval Head (Enc-self-1.2; Figure 9c)** Next, we looked for critical encoder heads that feeds to the RHS-Scanner Head and finally contributes to the Output Head's $Q$. Unlike the $K$-circuit discovery, where "keep-only-one-head" ablations is sufficient, multiple heads appear to contribute partial but complementary information. To isolate their roles, we measured drops in the output head's accuracy when ablating each encoder head individually while keeping the others intact (the "ablate-only-one-head" approach, more discussion in A.4).

This analysis highlighted **Enc-self-1.0** and **Enc-self-1.2** as the most critical (Figure 9a). In Enc-self-1.0, within the Support Set, each color token on the RHS attends back to its corresponding symbol on the LHS, inheriting that symbol's token and positional embedding (henceforth the Primitive-Retrieval Head; Enc-self-1.4 shows a similar but less dominant role) (Fig. 9b). Meanwhile, Enc-self-1.2 is similar, such that each color token on the RHS attends back to its function symbol on the LHS, passing that token and positional embedding on to the color token (henceforth the Function-Retrieval Head) (Fig. 9c).

Why do the color tokens on the RHS attend back to both kinds of information on the LHS? We reason that if a color token on the RHS were to encode it's primitive symbol's index-on-LHS: for example, in '...| D=pink | B S D=pink blue pink |...', pink were to encode 3rd inherited from D (D is 3rd in 'B S D'), the *absolute* position of D must be compared with the *absolute* position of the S to yield a *relative* position. Now that with the Primitive- and Function-Retrieval Heads, each RHS color token carries two positional references: (1) the associated LHS primitive, and (2) the function symbol, we hypothesize that by comparing these references, the model can infer the primitive symbols' index-on-LHS for each of the associated color tokens on the RHS. This computation is illustrated in Fig. 10a.

**Index-On-LHS Tracing** To confirm that our discovered circuit genuinely encodes the index-on-LHS in the Output Head's $Q$, we conducted three complementary ablation experiments summarized in Figure 10b:

- **Retaining only the Primitive- and Function-Retrieval Heads** When all other encoder heads are ablated, the RHS-Scanner Head's $Z$ still carries index-on-LHS that propagate to the Output Head's $Q$, indicating that these two heads alone provide sufficient index information.
- **Ablating the Primitive- or Function-Retrieval Head individually** Ablating either head disrupts the clustering by index-on-LHS in the RHS-Scanner Head's $Z$, demonstrating that both heads are necessary to preserve the full index information.
- **Ablating the RHS-Scanner Head (together with Dec-cross-0.0 and -0.3)** These decoder heads share similar attention patterns that track color tokens on the function's RHS. When all three are ablated, clusterings by index-on-LHS are eliminated from the Output Head's $Q$ (Figure 10c).

Thus, we conclude that the $Q$-circuit depends on the RHS-Scanner Head to capture the index-on-LHS information supplied by the Primitive- and Function-Retrieval Heads. By aligning these $Q$ signals with the $K$, the model consistently determines which token to generate next.

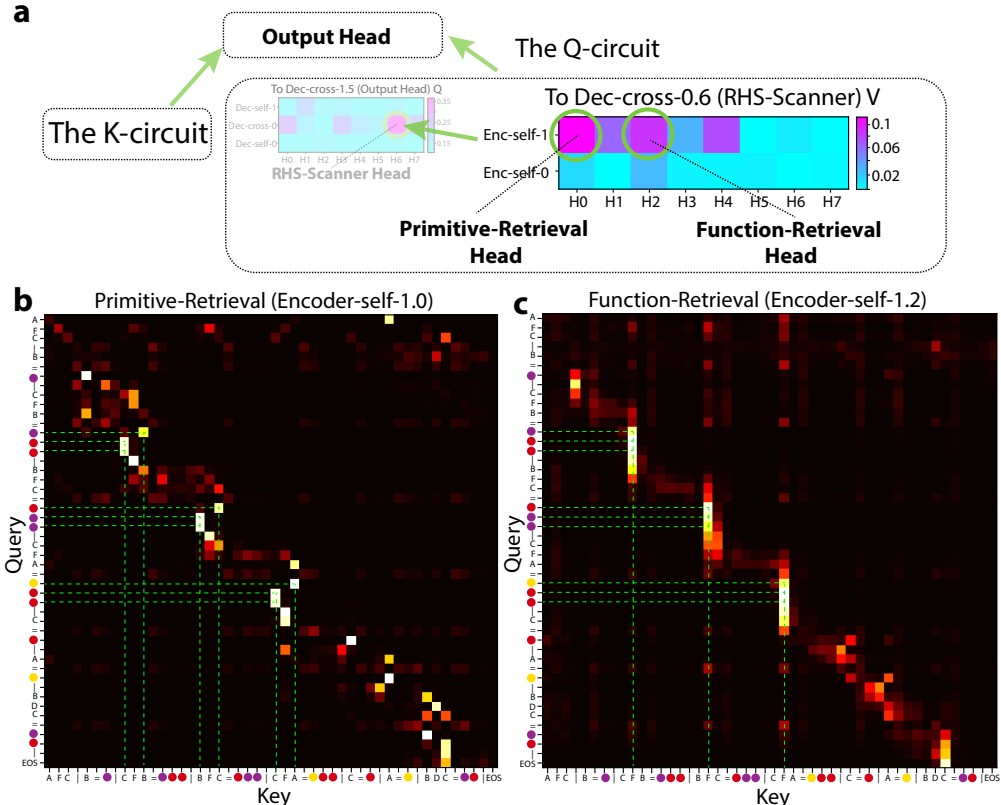

Figure 9: (a) Contribution to Output Head's performance via $Q$. (b) Contribution to Output Head's performance via the RHS-Scanner's $V$. (c) Attention pattern of Dec-cross-0.6. (d) and (e) Attention patterns of Enc-self-1.0 and Enc-self-1.2.

### 3.4   Targeted Perturbation Steers Output

Our circuit analysis indicates that the Output Head's $K$-circuit encodes the primitive symbols' *index-in-question*, while its $Q$-circuit encodes their *index-on-LHS*. If the model indeed relies on this positional indexing for token prediction, then perturbing (i.e., swapping) this index information should systematically alter the attention patterns and consequently the model's behavior.

**Perturbation Alters Attention Patterns**   Concretely, consider an input like 'A S B | A=red | B=blue | ...', where 'red' inherits 1st (from A) and 'blue' inherits 3rd (from B) in the $K$-circuit. If the $Q$-circuit expects a token with index value of 1st ('red' in this case), swapping these positional embeddings between A and B at the earliest node (Question-Broadcast Head's $V$) of the $K$-circuit—while freezing the $Q$-circuit—should revert the Output Head's attention from 'red' to 'blue' (Figure 11a).

Indeed, performing this targeted swap of positional embeddings caused the Output Head to shift its attention predictably from 'red' to 'blue', corresponding precisely to their swapped positions (Figure 11b). A control experiment using random positional embedding shuffles did not produce this systematic attention shift, confirming the causal role of positional indexing in the Output Head's $QK$ alignment.

**Perturbation Partially Alters Model Outputs**   We further assessed whether this targeted perturbation affects the final output logits. Post-perturbation, model accuracy dropped from **94%** to **76%**, remaining above chance but far below the original performance. Correspondingly, the logits for originally correct tokens decreased while those for swapped tokens increased (Figure 11c). The partial logit swap aligns with earlier observations (Figure 5), where Dec-cross-1.1 and -1.3 also provide significant contributions to final token logits. Again, random positional embedding shuffles did not replicate this systematic logit alteration.

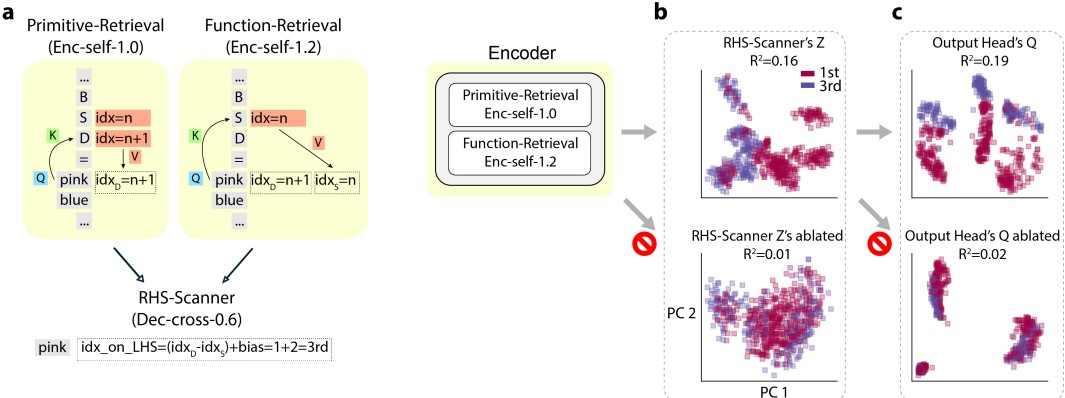

Figure 10: The computation pipeline for index-on-LHS. (a), In the *Primitive-Retrieval* head, the `pink` token retrieves the absolute position of the `D` token; in the *Function-Retrieval* head, the `pink` token retrieves the absolute position of the `S` token; then the *RHS-Scanner* head computes the difference of the two values to get the relative position of `D` on the function LHS. (b) and (c), Ablation results for token embeddings labeled by index-on-LHS. Concretely, for an episode with prompt 'A S B | A=red | B=blue | D=pink | B S D=pink blue pink | ...' and prediction 'SOS blue red blue EOS', in (b), points are the $Z$ of 'SOS' and 'blue' in the decoder input tokens (`SOS` is labeled `3rd`, because `SOS` attends to the `pink` on function RHS, and `D` is the `3rd` on the LHS; similarly, `blue` is labeled `1st`); in (c), points are the $Q$ of decoder input tokens (`SOS` is labeled `3rd`, `blue` is labeled `1st`). $R^2$ score quantifies the percentage of total variance explained by the index identity.

Together, these targeted perturbations confirm the causal role of the identified compositional circuit, demonstrating that precise manipulation of internal activations can systematically steer model outputs.

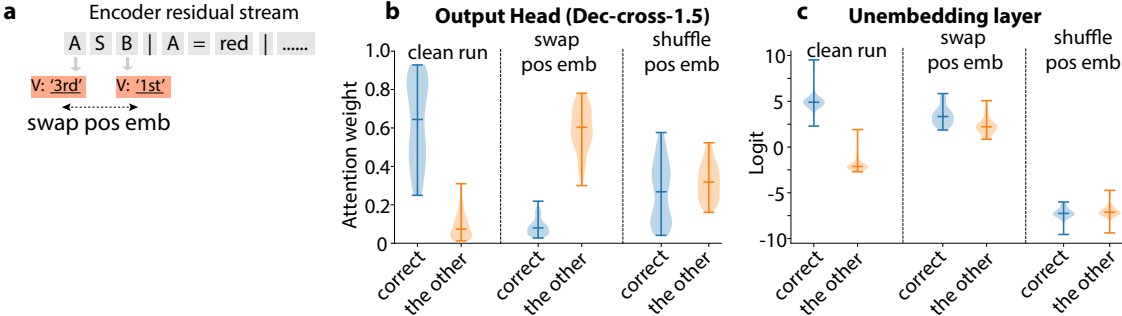

Figure 11: (a) Schematic illustrating the targeted swap of position embeddings. (b) Attention weights from the Output Head comparing the original correct token and swapped token across three conditions: unperturbed (left), targeted position swap (middle), and random shuffle control (right). (c) Similar comparison as (b), but for final output logits rather than attention weights.

## 3.5 No Linearly Decodable Function Vectors

Our task tests compositional generalization over functions. To check whether the model encodes explicit *function vectors* (as in Todd et al. (2023)), we probed whether the hidden states contain linearly decodable information about each function's identity. Using the full validation set, we trained linear probes on all residual streams and attention embeddings across all token positions.

Decoding accuracy remained at chance level, suggesting that no explicit function vector is stored. This supports our main finding that function tokens act as pointers for output remapping, and that generalization arises through a slot–content routing mechanism rather than dedicated semantic embeddings.

## 4    Discussion

In this work, we investigated how a compact transformer model performs compositional generalization on a synthetic sequence arithmetic task. Using *path-patching* and *causal ablations*, we uncovered a detailed $QK$ circuit that encodes index information from both the Question and the function's LHS. We further showed that precisely swapping these positional embeddings predictably alters the model's behavior, confirming the causal role of this circuit in supporting compositional generalization. These results demonstrate that even for complex functions, transformers can implement structured and interpretable mechanisms.

### 4.1    Limitations and Future Work

**Limited Model and Dataset.** We performed circuit interpretation on a single model instance. To test whether the discovered circuit mechanism is robust across different configurations, we trained an additional model with different hyperparameters and an expanded dataset. Specifically, we increased the number of encoder and decoder blocks from 2 to 3 and reduced the number of attention heads per layer from 8 to 4. For the dataset, we increased the number of arguments per function from 2 to a maximum of 5 and extended the maximum output length from 5 to 16 tokens. The new model achieves 96.0% accuracy on the test set. Applying the same circuit discovery approach, we identified a similar circuit in this new model, with all key functional attention heads replicated (see Appendix A.6). This result supports that the identified mechanism is not specific to a single architecture. However, determining whether similar interpretable mechanisms exist in large-scale, real-world LLMs remains an important open question for future work.

**Redundant Circuits.** Although our targeted activation edits successfully influenced the Output Head's behavior, they did not fully control the predicted tokens. This is due to the presence of redundant circuits with overlapping functions in the model (Lieberum et al., 2023; Conmy et al., 2023). We performed a head-by-head ablation experiment and found that removing any single head does not significantly degrade performance, indicating that each algorithmic step is supported by multiple functional heads. To enable precise output steering in larger models, it will be essential to develop systematic approaches for identifying and modifying all redundant pathways simultaneously.

**Specialized Compositional Generalization** The identified circuit is brittle and highly specialized to the sequence remapping tasks we defined. Indeed, simply repositioning the *function* symbol (e.g., 'A S B' to 'S A B') completely disrupts performance to chance level, which indicates that the model always assumes the *function* symbol to be on the 2nd position, lacking flexibility in symbol recognition. More flexible compositionality involving multiple or recursive functions clearly requires more complex circuit mechanisms, which remains an important direction for future exploration.

### 4.2    Broader Implications

Despite these limitations, **our study advances interpretability methods and deepens our understanding of compositional generalization in transformers.**

First, we present a rigorous, end-to-end circuit analysis methodology for a transformer solving a non-trivial compositional task. We demonstrate that such interpretability can enable targeted steering of model outputs through precise activation-level edits. This circuit-tracing workflow can serve as a template for future interpretability studies across different transformer architectures and tasks.

Second, we provide a new perspective on how 'functions' are instantiated in transformers. Through probing experiments, we confirm that the hidden embeddings do not encode the function semantics *per se*. Instead, the model's operation resembles Searle's "Chinese Room": it (1) leverages disentangled representations of token identity (content) and token position (slot), and (2) follows syntactic routing rules without forming an internal semantic concept of the function itself. This insight complements the 'function vector' view and demonstrates that transformers can generalize by exploiting structural routing rather than explicit function embeddings. It also generates testable hypotheses for other neural architectures—for example, whether similar routing mechanisms exist in recurrent networks or diffusion models when solving in-context learning problems.

We hope this study inspires further work on the mechanistic foundations of compositionality in large-scale models across broader task domains.

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

# A  Appendix

## A.1  Related Work

**Transformer circuit interpretation.** Mechanistic interpretability of transformers began with analysis of simplified models, identifying attention heads as modular components that implement specific functions.

In their seminal work, Elhage et al. (2021) and Olsson et al. (2022) introduced "induction heads" as critical components for in-context learning in small attention-only models. These heads perform pattern completion by attending to prior token sequences, forming the basis for later work on compositional generalization. Case studies have dissected transformer circuits for specific functions, such as the 'greater than' circuit (Hanna et al., 2023), the 'docstring' circuit (Heimersheim & Janiak, 2023), the 'indirect object' circuit (Wang et al., 2022), and the 'max of list' circuit (Hofstätter, 2023). These case studies successfully reverse-engineered the transformer into the minimal-algorithm responsible for the target behavior.

To facilitate identification of relevant circuits, researchers have proposed circuit discovery methods such as logit lens (nostalgebraist, 2020), path patching (Goldowsky-Dill et al., 2023), causal scrubbing LawrenceC et al. (2022). For large-scale transformers, automated circuit discovery methods are also proposed (Conmy et al., 2023; Hsu et al., 2024; Bhaskar et al., 2024). So far, transformer interpretability work still requires extensive human efforts in the loop for hypothesis generation and testing. We point to a review paper for a more comprehensive review (Rai et al., 2024).

**Compositional generalization in transformers.**  In their study, Hupkes et al. (2019) evaluated compositional generalization ability on different families of models, and found that transformers outperformed RNN and ConvNet in systematic generalization, i.e., recombination of known elements, but still incomparable to human performance. Zhang et al. (2024) pointed out that transformers struggle with composing recursive structures. Recently, Lake & Baroni (2023) showed that after being pre-trained with data generated by a 'meta-grammar', small transformers (less than 1 million parameters) can exhibit human-like compositional ability in novel in-context learning cases. This is in line with the success of commercial large language models (LLM) in solving complex out-of-distribution reasoning tasks (Bubeck et al., 2023; DeepSeek-AI et al., 2024), where compositional generalization is necessary.

Several studies highlighted factors that facilitate transformer's compositional ability. Wang & E (2024) identified initialization scales as a critical factor in determining whether models rely on memorization or rule-based reasoning for compositional tasks. Zhang et al. (2025) revealed that low-complexity circuits enable out-of-distribution generalization by condensing primitive-level rules. (Sanford et al., 2024) identified logarithmic depth as a key constraint for transformers to emulate computations within a sequence. Here, we offer a complementary mechanistic understanding of how transformers perform compositional computations.

## A.2  Transformer Model

We adopt an encoder-decoder architecture, which naturally fits the task by allowing the encoder to process the prompt (Question + Support) with bidirectional self-attention and the decoder to generate an output sequence with causal and cross-attention. Specific hyperparameters include:

- Token embedding dimension: $d_{\mathrm{model}} = 128$

- Attention embedding dimension: $d_{\mathrm{head}} = 16$

- Eight attention heads per layer (both encoder and decoder)

- Pre-LayerNorm (applied to attention/MLP modules) plus an additional LayerNorm at the encoder and decoder outputs

- Standard sinusoidal positional embeddings

The encoder comprises two layers of bidirectional self-attention + MLP, while the decoder comprises two layers of causal self-attention + cross-attention + MLP. We train the model by minimizing the cross-entropy loss (averaged over tokens) using the Adam optimizer. The learning rate is initialized at 0.001 with a warm-up phase over the first epoch, then linearly decays to 0.00005 over training. We apply dropout of 0.1 to both input embeddings and internal Transformer layers, and train with a batch size of 25 episodes. All experiments are performed on an NVIDIA A100 GPU.

### A.3 Task Structure

In each episode, the *Support Set* and *Question* are concatenated into a single prompt for the encoder, with question tokens placed at the start. Question, primitive assignments, and function assignments are separated by '|' tokens, while primitive and function assignments are identified by '='. Overall, there are 6 possible colors and 9 symbols that may serve as either color primitives or function symbols. Each episode contains 2–4 function assignments and 3–4 color assignments.

A function may be a single-argument (`arg func`) or double-argument (`arg1 func arg2`) function. The function's right-hand side (RHS) describes how arguments are transformed, generated by randomly sampling the arguments and interpreting them to color tokens. For example, we can define (`arg1 func arg2 = [arg2] [arg1]`) or (`arg1 func arg2 = [arg2] [arg1] [arg1] [arg2] [arg2]`) by sampling `arg1` and `arg2` certain times. The output length is randomly chosen between 2 and 5 for each function. Each prompt ends with an 'EOS' token. During decoding, the model begins with an 'SOS' token and iteratively appends each newly generated token until it emits 'EOS'.

We randomly generate 10,000 episodes for training and 2,000 for testing, ensuring that the primitive and function assignments in testing episodes do not overlap with those in the training set.

### A.4 Path Patching

Path patching is a method for isolating how a specific *source node* in the network influences a particular *target node*. It proceeds in three runs:

1. **Clean Run:** Feed the input through the model normally and *cache* all intermediate activations (including those of the source and target nodes).

2. **Perturbed Run:** Freeze all direct paths into the target node using their cached activations from the clean run. For the *source node* alone, replace its cached activation with *mean-ablated* values. Record the new, perturbed activation at the target node.

3. **Evaluation Run:** Supply the target node with the perturbed activation from Step 2, then measure any resulting changes in the model's output. This quantifies how the source node's contribution (altered via mean-ablation) affects the target node's behavior.

**Chained Path Patching.** When analyzing circuits that span multiple nodes in sequence, we extend path patching in a *chain-like* manner. For instance, to evaluate a chain $A \rightarrow B \rightarrow C$:

- We first perform path patching on the sub-path $B \rightarrow C$ as usual.

- Next, to capture how $A$ specifically influences $B$, we isolate and record $A$'s effect on $B$ via mean-ablation on all other inputs to $B$.

- Finally, we patch that recorded activation into $B$ and evaluate its effect on $C$.

For a chain of length $N$, we run $N + 1$ forward passes, ensuring the measured impact on the target node reflects only the chained pathway. This approach precisely attributes the model's behavior to the intended sequence of dependencies.

**Two Modes of Ablation.** To assess how individual heads or nodes contribute to the target node, we use two complementary modes:

1. **Keep-only-one-head:** Mean-ablate all direct paths to the target node except for *one* node, which retains its clean-run activation.

2. **Ablate-only-one-head:** Keep all direct paths to the target except one node, which is mean-ablated.

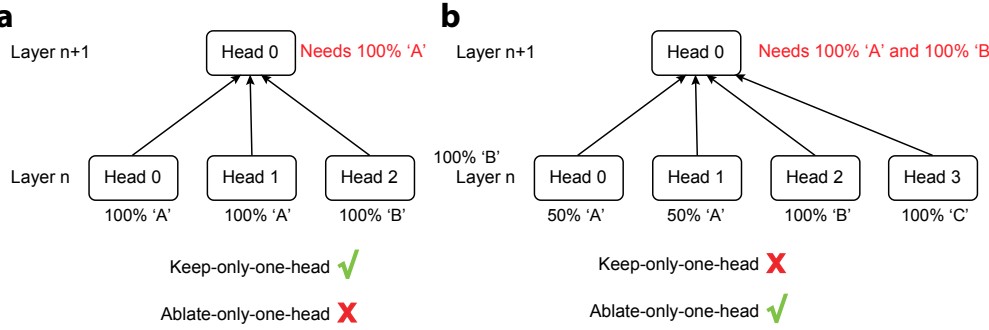

Figure 12: Two modes of ablation.

We use an illustration to show when and why each ablation mode are used (Figure. 12).

In case (a), suppose the downstream *Head 0 (Layer n+1)* needs information *A*. In *Layer n*, multiple copies of *A* are provided. In this case, *keep-only-one-head* can identify which heads gives *A*. In contrast, *ablate-only-one-head* will not be able to identify which heads give *A*, because *A* is always provided by other heads that saturates the performance.

In case (b), suppose the downstream *Head 0 (Layer n+1)* needs both information *A* and *B*. In this case, *keep-only-one-head* will not be able to identify the contributing nodes, because none of the nodes alone can support the function of the downstream node. Instead, *ablate-only-one-head* will identify the nodes contributing necessary information to the downstream node (Figure 10).

Our circuit identification uses *keep-only-one-head* ablation by default and switches to *ablate-only-one-head* if the former fails to find the dominant head. By combining both modes, we identified the putative QK-circuit of the output head. We then validate the circuits by inspecting the information they propagates and causally erasing the information by ablating specific upstream nodes.

## A.5 $R^2$ Score

To quantify how much an activation dataset $\mathbf{Y}$ encodes a particular latent variable $\mathbf{Z}$, we compute a linear regression of $\mathbf{Z}$ (one-hot encoded) onto $\mathbf{Y}$ and measure the explained variance:

$$R^2 = 1 - \frac{SS_{res}}{SS_{total}}.$$

An $R^2$ value of 1.0 indicates that $\mathbf{Z}$ fully explains the variance in $\mathbf{Y}$, whereas an $R^2$ near 0.0 implies $\mathbf{Z}$ provides no information about $\mathbf{Y}$.

## A.6 Replication with Different Architecture

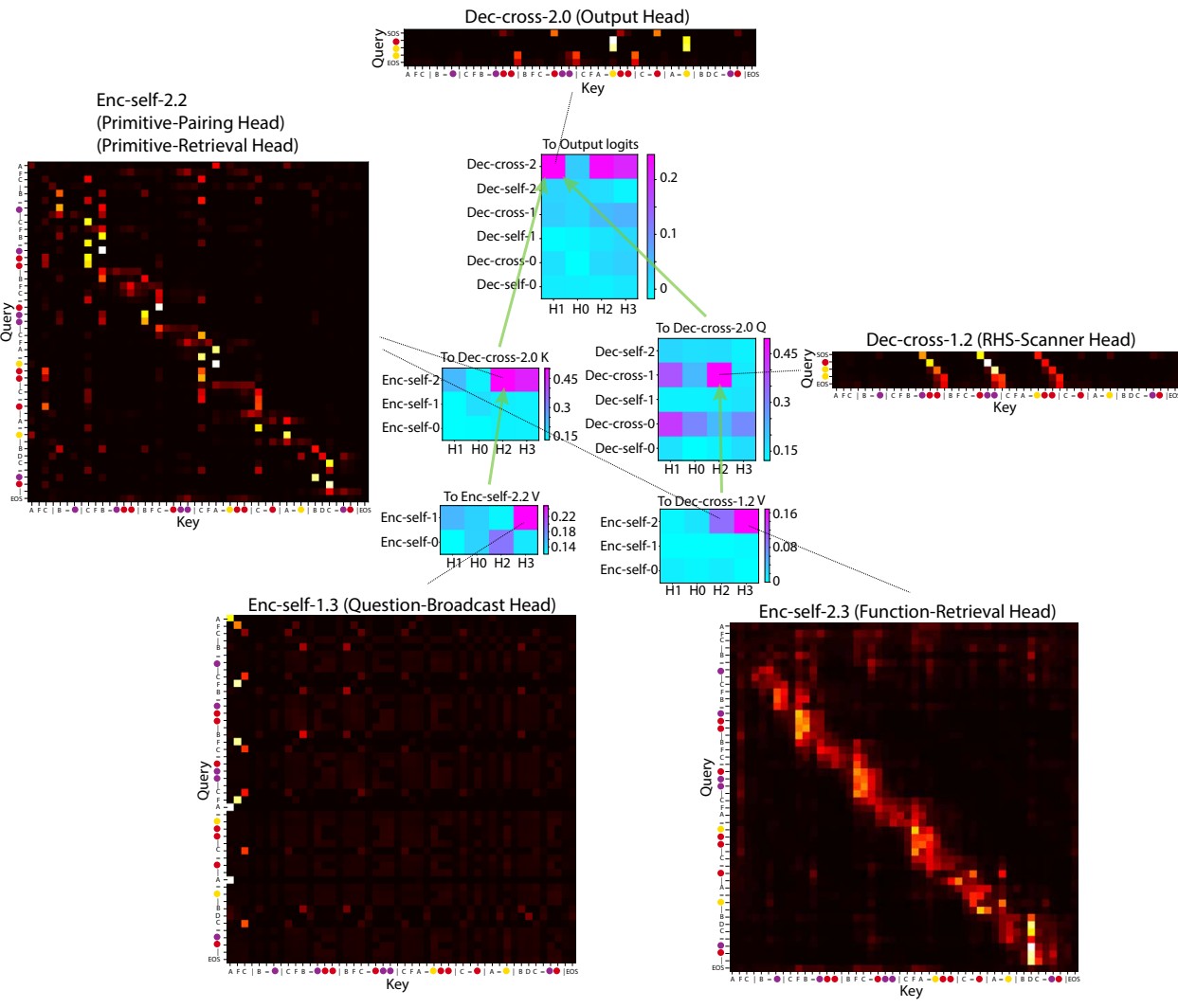

Figure 13: Circuit discovery on a different architecture. The model consists of 3 encoder/decoder layers with 4 attention heads in each layer. All the important attention heads are re-discovered in the new model, with the only exception that the original Primitive-Pairing and Primitive-Retrieval Heads are now merged as a single head. We speculate that this is due to their similar functions and the reduced number of heads in each layer.

