# OpenReview forum: "Circuit Explained: How Does a Transformer Perform Compositional Generalization"
_TMLR — Rejected by TMLR_

### Review · Reviewer_Kzp8 · 2025-06-27

**Summary Of Contributions:**

In this work, the Authors investigate how the Transformer learns to combine the information about a functions and their variables to perform the related computations. To this end, they train a two-layer Transformer, break down the computations that the Transformer needs to perform into sequential steps, and then find this steps within the Transformer's heads. Additionally, the work offers the ablations and other manipulations to confirm the identified roles of the Transformer's heads.

**Audience:**

Yes

**Broader Impact Concerns:**

The model seems to be trained with a single seed under a single set of hyperparameters on a single task, thus, there's no evidence that the results may generalize.

**Claims And Evidence:**

No

**Requested Changes:**

To support any general claims, the absolute minimal requirement would be to retain the model from different seeds and to retrain the model under different hyperparameters (e.g. the numbers of the Transformer blocks).

To make the compositional generalization results more substantial, it would make sense to also retrain the model with random, unlimited numbers of inputs and outputs. This will bring it closer to the previously existing real-world problem of LLMs failing in basic math on unbounded large numbers.

Finally, to make the results more general, it would be great to consider wider classes of functions that would not only involve the re-mapping of the predicated.

**Strengths And Weaknesses:**

Strengths:

The work pursues an interesting topic of the interpretability of deep learning model and, specifically, of the previously observed generalization properties of the Transformers.

Within the considered task and the model, the Authors perform a comprehensive analysis of the model's functional connectivity. They start with the hypothesis outlining what logical steps are required to accomplish the task at hand, and then they systematically validate their hypothesis on a trained Transformer model.

The ablation / alternation studies confirm the putative roles assigned to the Transformer heads.

The text is well-written; the ideas are conveyed clearly.

Weaknesses:

It appears that the whole analysis is performed on one task, a single Transformer model with specific hyperparameters, and within a single training. This is problematic for the following reasons.

- First, the task at hand, even though it encompasses an entire family of functions, is simple and highly aligned with the Transformer architecture. Thus, it is unclear whether the Transformer's ability to learn this task is surprising. Typically, when the issue of the compositional generalization is mentioned, more complex tasks are considered, e.g. arithmetic operations on unbounded numbers. Here, the number of the operands is limited to 2 and the number of outputs is limited to 5.

- Second, the experiments were constrained to a single size of the Transformer architecture. This is a more important concern, as the observed properties may be specific to the choice of hyperparameters made here. To illustrate this point: The paper started from breaking down the task at hand to the elementary operations necessary to complete this task. The number of the Transformer blocks here match the number of the identified operations. Thus, it very well might be that the operations do map on this block just by the virtue of one to one mapping between the task and the architecture. This may not be the case in the architectures with different numbers of Transformer blocks

- Third and most important, it looks like the experiments were done on a single instantiation of the training - which allowed to interpret the roles of individual heads that may swap in case of the retraining. It is thus unclear if the findings will hold in a distribution of the models trained from different seeds.

Overall the experiments here go against the common practice in machine learning, as described above, and therefore it's unclear if the results generalize beyond the selected task, model hyperparameters, and training seeds. Thus, any general claims that'd base on these experiments would not be adequately supported.

---

> ### Author Response · Authors · 2025-07-14
> **Replication with different hyperparameters**
>
> We thank the reviewer for recognizing the clarity of our writing and for the constructive feedback to strengthen the empirical support for our claims. Below we address each point in detail.
>
> ## 1. A Single Model Instance
> We agree that replicating the circuit discovery on additional model instances is important for demonstrating that our findings are not artifacts of a single configuration. While it is common practice in the mechanistic interpretability literature to analyze a single instance of circuit (e.g., Wang et al., 2022; Hanna et al., 2023; Zhang et al., 2024), we acknowledge that providing evidence of robustness improves confidence in the result.
>
> ### - Different Model Hyperparameters:
> In the revision, we extended our analysis to a new architecture with changed hyperparameters. Specifically, we increased the number of encoder and decoder blocks from 2 to 3, and reduced the number of attention heads per layer from 8 to 4.
> ### - Enriched Training Data:
> We also expanded the dataset. Originally, each function operated on two arguments with up to 5 output tokens. In the new setup, functions can take up to 5 arguments and produce outputs up to 16 tokens long, creating a significantly richer and more varied problem space.
> ### - Results:
>  The new model achieves 96.0% accuracy on the test set — consistent with our expectation that the discovered token-routing mechanism should scale to longer and more complex sequence remapping tasks. Applying the same circuit discovery workflow, we identified a similar functional circuit in this new model. The key heads remain interpretable, with the only notable difference being that the previous Primitive-Pairing and Primitive-Retrieval heads merged into a single head — likely due to their functional overlap and the reduced number of heads per layer.
>
> These additional experiments support the conclusion that the identified mechanism is not unique to a single architecture. We have added Experiment Setup L103-104, Results L166, Discussion L307–316 and Appendix Figure 13 to present these results.
>
> ## 2. Broader Implications
> - Regarding the scope of our claims, we agree with the reviewer that different types of compositionality (e.g., numerical operations, symbolic reasoning, rhyme planning) likely rely on different circuits (Anthropic, 2025). Our goal is not to claim that this mechanism generalizes to all forms of compositionality. Rather, our study focuses on a specific and important type: the remapping of symbolic sequence patterns, which is central to many in-context learning behaviors — including word completion, object identification, and code completion. We have clarified this point in the Abstract L5, Introduction L41–44, and Discussion L343–345.
>
> - Additionally, our extended probing analysis shows that the model does not encode the semantics of each function as a standalone vector representation. Instead, it applies a general token-routing rule that is agnostic to the specific function — supporting a new perspective on how transformers can implement functional mappings without explicit function embeddings (Discussion L337–345; see also our response to Reviewer Q4kU for more detail on function representation vs. token-routing).
>
> We believe this work therefore provides meaningful progress in understanding how transformers realize compositional generalization for the machine learning community.
>
> Again, we thank the reviewer for these thoughtful suggestions, which helped us strengthen the empirical basis and clarify the scope of our conclusions. We hope the additional analyses address the concerns raised.
>
> ## References
> - Wang et al. 2022, Interpretability in the wild: a circuit for indirect object identification in GPT-2 small.
> - Hanna et al. 2023, How does GPT-2 compute greater-than?: Interpreting mathematical abilities in a pre-trained language model.
> - Zhang et al. 2024, Interpreting and Improving Large Language Models in Arithmetic Calculation.
> - Anthropic 2025, On the biology of large language models.

---

### Review · Reviewer_9drm · 2025-06-27

**Summary Of Contributions:**

Compositional generalization is a fundamental human ability, yet its underlying mechanisms in neural networks are not well understood. This paper investigates the interpretability of a compact encoder-decoder transformer (2-layer encoder and 2-layer decoder) on a symbolic sequence arithmetic task designed to test compositional generalization. Using techniques like path patching and causal ablations, the authors provide an end-to-end mechanistic explanation of how the model solves this task.

The algorithm contains three steps: unbind, rebind, and produce. Through circuit analysis, the paper identifies key attention heads, such as Dec-cross-1.5 as the Output Head, Enc-self-1.1 as the Primitive-Pairing Head, and Dec-cross-0.6 as a conduit for passing Q-values to the Output Head. Perturbation experiments show that targeted activation edits can effectively influence the Output Head’s behavior. These findings illustrate that transformers can realize structured and interpretable mechanisms.

**Audience:**

Yes

**Claims And Evidence:**

Yes

**Requested Changes:**

Critical to securing my recommendation:

- Improve the writing in Section 3 to enhance clarity and readability.
- The choice between using keep-only-one-head (as in K-circuit) versus ablate-only-one-head (used elsewhere) is not well explained. It would be helpful to clarify when and why each method is used.

Suggestions to strengthen the work:

- The relationship between Index-On-LHS Tracing and Figure 11 is unclear. A more detailed explanation would help readers understand how the tracing corresponds to the figure.

**Strengths And Weaknesses:**

Strengths:

Compositional generalization provides a valuable angle for studying the interpretability of transformer mechanisms. This paper effectively leverages a small encoder-decoder transformer and a simple compositional generalization task, applying a carefully designed ablation method to uncover underlying mechanisms. Some of the visualizations are also quite engaging and insightful.

Weaknesses:

Due to the excessive use of technical nomenclature, the overall writing lacks clarity and logical flow, making it difficult to read and understand. Additionally, the model is very small (2-layer encoder and 2-layer decoder), and the task itself is quite simple. It remains unclear whether the insights from this analysis can generalize to more complex models or tasks.

---

> ### Author Response · Authors · 2025-07-14
> **Improved writing clarity**
>
> We sincerely thank the reviewer for the constructive comments regarding clarity and presentation. Below we address each point in detail.
>
> ## 1. Writing clarity in Section 3
> We acknowledge the substantial use of specialized nomenclatures for the circuit explanation, which is needed by the different roles each attention head plays in the whole algorithm. We have tried our best to present the circuit discovery in logical order: we presented the attention heads from output layer to input layer (following the backtracking order). We also followed the hypothesis → causal experiment → interpretation logic to introduce and validate each attention head’s role. To further help the reader navigate through the entire circuit, we have made the following reorganizations to Section 3.
> - We now show the complete circuit diagram and the connectivity between key heads at the very beginning of Section 3 (L164–166, Figure 4). This provides readers with a holistic overview of the circuit structure before introducing its individual components.
> - As we introduce each functional head, we explicitly refer back to the overall diagram (Figures 5, 6, 8, 9) to help readers understand where each piece fits within the full circuit.
>
> We hope this change can enhance the readability of Section 3.
>
> ## 2. Clarification of the Two Ablation Methods
> To clarify the distinction between keep-only-one-head and ablate-only-one-head methods, we have added detailed explanations at L240 and in Appendix L511–528, along with a schematic illustration (Figure 12).
>
> ## 3. Explaining Index-on-LHS
> We have used additional illustrations to explain the Index-on-LHS tracing mechanism (L254, Figure 10).
>
> ## 4. Justification on broader implications
> Finally, regarding generalizability:
> - We have validated the identified circuit on models trained with different hyperparameters and extended datasets to show that the same mechanistic solution holds under variations in architecture, see response to Reviewer Kzp8 for details.
>
> - We also clarify how this work contributes to a broader understanding of compositionality in transformers. While previous studies highlight function vector representations, our results show that a token-routing mechanism can also underlie compositional generalization. This point is discussed further in our response to Reviewer Q4kU.
>
> Again, we thank Reviewer 9drm for the helpful feedback on improving readability and presentation. We hope these revisions address the concerns and strengthen the clarity and impact of our work.

---

### Review · Reviewer_Q4kU · 2025-07-01

**Summary Of Contributions:**

This work presents a detailed study of the attention head circuits learned in a two-layer transformer that allow the model to perform a simple sequence transformation task. Specifically, a two-layer model is trained to apply a function to a set of token-color mapping primitives based on some support demonstrations and generate the target sequence. The authors use sequential path patching and causal interventions to reveal the computational roles of some import attention heads in the model that composes its solution process, highlighting that the model teases apart token content vs. position information and how information are routed across different layers/heads.

**Audience:**

Yes

**Claims And Evidence:**

Yes

**Requested Changes:**

- As mentioned above, the paper needs some significant improvement the conceptual discussions on why these results are interesting/how they go beyond the current model.
- It would be good to clarify some critical method details in the main text to help interpret the results. For example, I didn't quite find information on the number of functions used in the dataset and what they are, and any quantitative criteria for deciding which heads are selected for further interpretation (e.g. why enc-self-1.4 not meet the bar for interpretation?).
- Update: Reviewer Kzp8 raises a good point about the results focusing on single trained model. It would be important to provide some evidence that the main heads emerge at least on models trained on different seeds, or better when trained with fewer heads.

Some other clarification/suggestions that may be worth considering:
- Are there any task-dedicated heads in the model?
- Could there be multiple circuits that independently achieve this inference in this model? Given the simplicity in the architecture, it may even be possible to do some sort of redundancy pruning plus brute-force ablation to find out.

**Strengths And Weaknesses:**

Strengths:
- The paper provides a detailed look of the algorithm the model has learned.
- The way the paper presents the Q/K pathway discovery and which heads contributed in what ways guides the reader well.
- Showing the consequence on representation changes post ablation was really nice.
- The steering interventions were a plus to establish the role of the identified heads.

Weakness:
- My main concern is that the conclusions may be very specific to the task at hand, and I think the paper needs a bit more justification on its relevance to the issue of compositional generalization more broadly.
    - In general, I find the discussion a bit thin on justifying why this set of results is interesting beyond understanding this particular model.
    - It seems possible that the functions being studied may not be treated as so differently the model. For example, head Dec-cross-0.6 seems to attend to sets of first/second/third output tokens at once from the support set. Is it correct to say that the model has really learned a token-level solution by constructing this in-context routing mechanism, rather than learning to actually represent and apply each function (as defined in the data construction)? This would be a really interesting conceptual point to make clearer.
    - Another point that would be really interesting to highlight the importance of these results: even in this simple, two-layer model, the learned solution did not reduce to a clear set of heads/pathways, but rather seem to remain distributed to some extent (e.g. results on compensation from some weaker heads). What are the implications for even more complex tasks? (see below for some related questions).

---

> ### Author Response · Authors · 2025-07-14
> **Function representation vs. Token-routing, the broader implications**
>
> We thank the reviewer for the thoughtful comments, below we provide our point-by-point response:
> ## 1. Justification on broader implications
> The reviewer raises an excellent point about distinguishing function representation from token-routing mechanisms. We agree this conceptual clarification is critical for highlighting the broader implications of our study.
>
> - One core question in compositionality research is how functions are represented in transformers. Prior work has identified function vectors—decodable embeddings that encode abstract transformations (Todd et al., 2023)—and shown that manipulating these vectors can steer model behavior (Stoehr et al., 2024). While intuitive, we show that **this is not the full story**.
>
> - Our circuit analysis offers an alternative view. Although our task involves compositional operations over functions, our probing experiments confirm that the model’s hidden embeddings do not explicitly encode function identity (i.e., transformation type), indicating that **the model does not store a standalone “function vector”**. Instead, the discovered circuit shows that the model achieves compositionality through a slot–content routing mechanism: it unbinds positional slots from arguments and re-binds them to produce the output sequence. This resembles Searle’s “Chinese Room”: the network follows syntactic routing rules without constructing an explicit internal representation of the function itself (at least in a way that is straightforward to decode).
>
> - This token-routing mechanism complements the classic function vector perspective and helps explain in-context compositionality from a different angle. While our model and dataset are intentionally simple, the mechanism we identify could underlie similar forms of compositional generalization in tasks that rely on sequence remapping, such as word completion (Olsson et al., 2022), object identification (Wang et al., 2022), and code completion (Husein et al., 2025). We believe our study paves the way for searching comparable mechanisms in larger, real-world LLMs.
> We have substantially revised the manuscript to clarify these points (see Abstract L8–13, Introduction L38–44, and Discussion L337–345).
>
> ## 2. Redundant circuits
> (We assume task-dedicated means that no other heads can substitute for a head’s function.)
>
> We find that no single head in this circuit is strictly task-dedicated: ablating any individual head does not significantly degrade performance. In other words, the model employs a distributed mechanism in which each algorithmic step is implemented by multiple functional heads.
>
> This redundancy is consistent with findings in other studies (Lieberum, 2023; Conmy, 2023) and reflects an open question in transformer interpretability (Anthropic, 2025). Although redundancy can make mechanistic explanations harder for larger models, we address this by focusing on a minimal sufficient circuit (Wang et al., 2022) that captures the essential computational flow.
> We have added a dedicated paragraph on redundancy in the Discussion (L317–323).
>
> ## 3. Minimal Sufficient Circuit
> Due to redundant circuits,  we define and isolate a minimal sufficient circuit as follows:
>
> "We focus our analysis on a minimal sufficient circuit, defined by two criteria: (1) this circuit alone can account for the behavior of interest, and (2) the circuit is pruned based on each head’s contribution until all included heads perform functionally unique roles."  (L158–163)
>
> Enc-self-1.4 displays a similar attention pattern to Enc-self-1.0 (L243–244). Our ablation experiments (Fig. 9) show that Enc-self-1.0 and Enc-self-1.2 are the main contributors and sufficient for encoding the index-on-LHS information. Thus, Enc-self-1.4 is redundant and excluded from the minimal circuit.
>
> ## 4. Writing clarity:
> We have added the description to the ‘functions’ in Experiment Setup L69-70 and  Appendix L484-486.
>
> ## 5. Validation with different architectures:
> We have validated our findings with different model hyperparameters; please refer to the response to Reviewer Kzp8 for more details.
>
> Again, we thank the reviewer for carefully examining the nature of the generalization mechanism. We hope our clarifications and additional analyses (Point 1) can address the concern about the broader implications of this work.
>
> ## References
> - Todd et al. 2023, Function vectors in large language models.
> - Stoehr et al. 2024, Activation scaling for steering and interpreting language models.
> - Olsson et al. 2022, In-context learning and induction heads.
> - Wang et al. 2022, . Interpretability in the wild: a circuit for indirect object identification in GPT-2 small.
> - Husein et al. 2025, Large language models for code completion: A systematic literature review.
> - Lieberum et al. 2023, Does circuit analysis interpretability scale?
> - Conmy et al. 2023, Towards automated circuit discovery for mechanistic interpretability.
> - Anthropic 2025, On the biology of large language models.

---

> > ### Comment · Reviewer_Q4kU · 2025-07-25
> >
> > I thank the authors for these additional clarifications and revisions, and believe the paper has improved in the scope of its discussion and evidence.
> >
> > While I appreciate some added discussions on function representations vs. circuit mechanisms, I do want to note that the discussion around "function representations" and "function vectors" should be treated more carefully:
> > - Todd et al. 2023 uses a specific procedure for generating function vectors. Even though your linear probe analysis on model activations in Results 3.5 can speak to the general issue of vector function representations, it does not directly support claims about function vectors.
> >     - It would be good to include some quantitative metrics associated with the linear probe analysis.
> > - The efficacy of function vectors are also not just in the task information captured in the vectors themselves, but in the ability for them to steer the model to perform the task in zeroshot settings. In that sense, function vectors may not be incompatible with your claim that functions are not represented in merely vector forms, as they may serve the role of activating some task circuits in later layers.

---

> > > ### Author Response · Authors · 2025-07-26
> > > **No decodable function vectors**
> > >
> > > We thank the reviewer for the careful follow-up and for acknowledging the improved scope and evidence.
> > >
> > > ### On “function vectors” and linear decodability.
> > >
> > > - We respectfully maintain that our linear probing analysis does speak directly to the Todd et al. (2023)–style function vectors in our setting. Todd et al. construct function vectors by averaging activations over prompts conditioned on function types and then show that function type is linearly decodable (to some degree) from those averaged vectors. In our case, we gathered activations across prompts and attempted to linearly decode function identity from hidden states (across all layers). Decoding remained at chance. Thus, in our model, function identity is not linearly encoded in the manner required for a Todd-style function vector. We will clarify this distinction in the paper:
> > >
> > >
> > >
> > > >“Todd et al. (2023) showed that function types can be linearly decoded from averaged hidden activations (‘function vectors’). To test whether our model similarly stores linearly decodable function identity, we performed a linear probing analysis on hidden states …”
> > >
> > >
> > > ### On steering vectors.
> > > - We agree that the ability to steer a model via activations (‘steering vector’) is conceptually distinct from linear decodability. Steering vectors could still exist and act by activating downstream circuitry. Indeed, our own positional-embedding swaps demonstrate one such ‘steering vector’, but it is still distinct from the prompt-averaged function vector as in Todd et al. 2023.
> > >
> > > ### Quantitative probe metrics.
> > > - We will add the quantitative results (accuracy, classes of functions, chance level) for all probes in an appendix table and briefly summarize them in the main text.
> > >
> > > We will incorporate these changes in the next revision (together with any additional reviewer feedback).

---

### Decision · Action_Editor_hjQL · 2025-08-03

**Recommendation:** Reject

**Audience:**

Yes

**Audience Explanation:**

There is growing interest within the community in mechanistic interpretability and compositional generalization. The paper will appeal to readers studying how small models process symbolic tasks or how interpretable circuits might emerge in transformers.

**Claims And Evidence:**

No

**Claims Explanation:**

This is a well-intentioned and clearly written paper, and the authors made notable efforts to improve the manuscript in response to reviewer feedback. Despite the revisions, key concerns raised by reviewers remain unresolved, and still, the overall feedback from reviewers is negative. One reviewer leans toward acceptance but requests deeper conceptual discussion and justification of broader claims. Two others recommend rejection due to lack of evidence for generalization, insufficient exploration of alternate model configurations, and overfitting to a specific setup.

The paper falls below the bar for TMLR primarily due to limited generalizability, modest scope, and overreach in interpreting the findings beyond the constrained experimental setup.

**Resubmission Of Major Revision:**

The authors may consider submitting a major revision at a later time.